# Associations between Nutrients and Foot Ulceration in Diabetes: A Systematic Review

**DOI:** 10.3390/nu13082576

**Published:** 2021-07-27

**Authors:** Nada Bechara, Jenny E. Gunton, Victoria Flood, Tien-Ming Hng, Clare McGloin

**Affiliations:** 1Department of Diabetes and Endocrinology, Blacktown-Mt Druitt Hospital, Blacktown, NSW 2148, Australia; Nada.Bechara@health.nsw.gov.au (N.B.); Tien-Ming.Hng@health.nsw.gov.au (T.-M.H.); Clare.McGloin@health.nsw.gov.au (C.M.); 2Westmead Hospital, Sydney Medical School, Faculty of Medicine and Health, The University of Sydney, Westmead, NSW 2145, Australia; 3Centre for Diabetes, Obesity and Endocrinology Research (CDOER), The Westmead Institute for Medical Research, The University of Sydney, Westmead, NSW 2145, Australia; 4Garvan Institute of Medical Research, Darlinghurst, NSW 2010, Australia; 5Westmead Hospital, Research and Education Network, Western Sydney Local Health District, Westmead, NSW 2145, Australia; Vicki.Flood@sydney.edu.au; 6Faculty of Medicine and Health, Sydney School of Health Sciences, The University of Sydney, Camperdown, NSW 2006, Australia

**Keywords:** foot ulcer, nutrition, vitamin deficiency, wound healing

## Abstract

We reviewed the literature to evaluate potential associations between vitamins, nutrients, nutritional status or nutritional interventions and presence or healing of foot ulceration in diabetes. Embase, Medline, PubMed, and the Cochrane Library were searched for studies published prior to September 2020. We assessed eligible studies for the association between nutritional status or interventions and foot ulcers. Fifteen studies met the inclusion criteria and were included in this review. Overall, there is a correlation between poor nutritional status and the presence of foot ulceration or a delay in healing. However, there is not enough data to reach conclusions about whether the relationships are causal or only association. Further research is required to test whether any forms of nutritional supplementation improve foot ulcer healing.

## 1. Introduction

Diabetic foot ulcers (DFU) are a severe, yet unfortunately common, complication of diabetes which cause substantial morbidity. They are also associated with high mortality risk, which is similar to overall five-year cancer mortality [1]. It is estimated that 19–34% of people with diabetes will develop a foot ulcer in their lifetime, and many of these will result in a digit or lower limb amputation [2]. Every year in Australia, there are 10,000 hospital admissions for DFU, with approximately 4400 of these resulting in an amputation [3,4]. This has large costs to the health care system. Research suggests that investing in evidence-based practice for DFU could save the Australian government and health-care system around $2.7 billion over five years [5].

It is estimated that 85% of these diabetes related amputations could be prevented if wounds were detected early enough and managed appropriately [6]. Prevention of lower limb amputation would have significant positive effect on quality of life (QoL) of the individual and would be cost-effective.

The International Working Group on the Diabetic Foot (IWGDF) 2019 guidelines state on page 164, “do not use interventions aimed at correcting the nutritional status (including supplementation of protein, vitamin and trace elements, pharmacotherapy with agents promoting angiogenesis) of patients with a diabetic foot ulcer, with the aim of improving healing in preference to best standard of care”. However, it is known that certain vitamins and minerals play important roles in wound healing [7]] and if people are deficient in those factors, their wound healing would be impaired. 

Deficiency of vitamin C in its severest form causes scurvy. Symptoms can include bleeding gums, petechiae, arthralgia, tiredness, delayed wound healing, and an ability to bruise easily [8]. Vitamin C deficiency results in defective formation of connective tissues and collagen [8]. Vitamin C is a co-factor in lysine and proline hydroxylation on collagen molecules and hydroxylation of these amino acids is essential in stabilising the triple-helix structure of mature collagen [7]. This is especially important in the proliferation phase of wound healing when fibroblasts produce collagen which forms the scaffolding structure to support healing. The hydroxylation also provides essential tensile strength to the newly formed collagen, which would otherwise not stretch without tearing [9]. Tensile strength is important in healed wounds; if it is not sufficient, wounds may break down, leading to re-ulceration [2]. 

Vitamin C also plays a critical role in immune function which needs to be taken into consideration in foot ulcers because of the break in the epithelial barrier [9]. It is required for innate immune function, including the neutrophil ‘oxidative burst’ which is important for early bacterial killing. Anderson [2005] advocates that vitamin C is the most essential micronutrient in wound healing due to its role in collagen synthesis and angiogenesis [9].

Vitamin A is an essential fat soluble vitamin which has numerous bodily functions. Dietary vitamin A is absorbed as retinol from either preformed retinoids or provitamin A carotenoids [10]. Retinoids play an important role in wound healing as it is involved in cell mitosis, angiogenesis, increasing epithelial thickness, and collagen synthesis [11].

Zubair et al. reported that lower concentrations of vitamin D played a role in the pathogenesis of DFU and a 2019 meta-analysis reported that severe vitamin D deficiency is associated with the pathogenesis of foot ulceration [12,13]. Vitamin D is important for normal skin function; mice with deletion of the vitamin D receptor having a number of skin problems and poor wound healing [14]. 

Minerals such as zinc, copper, and selenium have roles in protein and collagen synthesis and cell replication, as well as antioxidant properties [15]. Zinc deficiency affects all phases of wound healing and causes a reduction in mature B-cells. Copper assists in the formation of red blood cells and aids vitamin C to form elastin [15]. Selenium is involved in antioxidant function and may protect the body from infection [16].

This paper will review published data about nutritional deficiencies or supplementations and healing of DFU.

## 2. Materials and Methods

The following databases were searched to identify relevant studies for inclusion:(i)Cochrane Library (searched 17 September 2020)(ii)Ovid Embase (searched 17 September 2020)(iii)Ovid MEDLINE (searched 17 September 2020)(iv)PubMed (searched 17 September 2020)

The combination of search terms used was as follows: (vitamin OR nutrient OR supplement) AND (foot OR feet) AND (wound OR ulcer OR ulceration). For PubMed, the search terms used were MeSH (Foot ulcer) and nutri*.

No limitations were placed on date of publication or language. Titles and abstracts were reviewed, and full copies were obtained for those potentially eligible for inclusion.

One author (N.B.) screened the search results. After duplicates were removed, titles and abstracts of the remaining papers were screened. The full manuscripts for those considered potentially relevant were read. The reference lists for these papers were also examined for potential studies to be included in this review resulting in the retrieval of one further study for full text reading. The search process is outlined in Figure 1.

### Inclusion and Exclusion Criteria

Studies had to meet the following criteria for inclusion: (a) randomised controlled trials, cohort studies or case-control studies; (b) analysis of the relationship between a nutrient or mineral, and foot ulceration; (c) report on serum concentrations or deficiency rates of the nutrient or mineral being analysed; and (d) human studies. 

Systematic review articles and case reports were excluded. For the study to be included, there had to be at least one foot outcome reported.

## 3. Results

### 3.1. Study Selection

Following the database searches, a total of 366 articles were found. Of these, 303 papers were excluded after screening and 17 duplicates were removed, leaving 46 articles for abstract review. Eighteen full text articles were assessed along with their reference lists. One study was retrieved for full text reading from a reference list. Of these 19 articles, 15 were suitable for inclusion in the study. The other four were excluded because no outcome was reported (*n* = 3) or because they did not assess foot wounds (*n* = 1).

### 3.2. Included Studies

Nine studies were randomised controlled trials [17,18,19,20,21,22,23,24,25]. There were two case control studies [26,27]. Four were cohort studies; one was a retrospective study [28], and three were prospective [12,29,30]. The studies are described in Table 1.

All studies were published in English. Six studies were reported from Iran [17,19,20,21,24,25] all by the same group. Two studies were carried out in India [12,18], two in the USA [22,28], and one in each of the following; Australia [29], Bahrain [30], Nigeria [26], Saudi Arabia [27], and Sweden [23].

### 3.3. Demographic Data

The studies had a total of 1737 participants in the pooled sample; 929 were in the intervention or case groups, and 808 people were controls. Of the 808 in the control groups, 381 had diabetes and foot ulcers, 377 had diabetes without foot ulcers, and 50 did not have diabetes or foot ulceration. Thirteen studies included both males and females, one study included males only [28], and the remaining study did not report gender [26].

The characteristics of both groups of participants, those with and without foot disease, are summarised in Table 1.

Populations evaluated in the included studies were as follows: (i)Six RCTs, all from the same senior author in Iran, included only participants with grade 3 DFU according to the Wagner classification system [17,19,20,21,24,25];(ii)One RCT included participants with grade 2–3 DFU [18,23];(iii)One RCT included participants grade 1–2 DFU [23];(iv)The remaining RCT included participants with diabetes who had at least one University of Texas grade 1A foot ulcer [22];(v)One retrospective [28] and one prospective study [29] included those with diabetes and foot ulceration;(vi)One case-control [27] and one prospective study [12] included participants with diabetes and foot ulceration, and a control group with diabetes but no foot ulceration;(vii)The other case-control study included participants with diabetes and foot ulceration, and a control group without diabetes or foot ulceration [26];(viii)The remaining prospective cohort study included any participants with foot ulceration; 90% of whom had diabetes [30].

The foot ulcer classification systems are outlined in Table 2 and Table 3 and Figure 2.

#### 3.3.1. Vitamin C

Two articles met the inclusion criteria and reported vitamin C concentrations in individuals with foot ulceration [26,29] (Table 4).

A case-control study by Bolajoko et al. (2017) assessed vitamin C, vitamin E, and selected minerals in people with diabetes and foot ulceration. There were 70 individuals with DFU and 50 participants free of diabetes and foot ulceration. The researchers measured vitamin C, vitamin E, copper, selenium, zinc and oxidative stress markers including lipid peroxide, total antioxidant status, superoxide dismutase and glutathione peroxidase. Significantly lower vitamin C (3.8 vs. 5.6 μmol/L, *p* = 0.003), selenium (0.48 vs. 0.81 μmol/L, *p* < 0.001), vitamin E (19.6 vs. 25.6 μmol/L, *p* < 0.001) and total antioxidant status (0.67 vs. 1.42 mmol/L, *p* < 0.001) concentrations were found in the DFU group compared to controls. Unexpectedly, people with DFU had higher glutathione peroxidase. Vitamin C was significantly positively correlated with glutathione peroxidase. The vitamin C assay was ‘old school’ and it is noteworthy that everyone tested had severe deficiency by usual-assay-standards, including the people without diabetes or foot ulcers [32]. People with DFU had concentrations which were clearly lower at 3.76 vs. 5.57 μmol/L.

In the prospective cohort study by Pena et al. (2020) serum concentrations of vitamins A, C, and D, and copper and zinc were measured in patients with DFU [29]. Vitamin C deficiency (<11.4 μmol/L) affected 50.8% of patients, with suboptimal concentrations (between 11.4 and 22.7 μmol/L) in another 22.2%. Over half of the 131 participants had low or undetectable plasma vitamin C. It was concluded that increased severity of DFU, assessed using the WIfI score (Figure 2), was associated with lower concentrations of vitamin C (*p* = 0.02).

#### 3.3.2. Vitamin D

Four studies report on vitamin D as the primary measure [12,18,19,30]. Another study by Pena et al. (2020) also reported on vitamin D deficiency in people with DFU.

Kamble et al. (2020) conducted a RCT with 60 participants. Group A was prescribed a weekly 60,000 IU sachet of cholecalciferol, and group B received a placebo. The trial ran for a period of 12 weeks and vitamin D concentrations significantly increased (*p* = 0.0001). They reported significant improvements in HbA1c (*p* = 0.008) and cholesterol concentrations (*p* = 0.0001). The study reported beneficial effects on ulcer healing when compared to the control group (*p* = 0.0001). This is one of the 6 studies from the Aseni group in Iran. We note the six studies have improbable homogeneity of recruited subjects and benefits from the six treatments (Table 5).

Razzaghi et al. (2017), also from the Aseni group completed a study of very similar design with 60 participants. The intervention group received 50,000 IU of vitamin D every 2 weeks over 12 weeks, and controls received placebo. The patients had diabetes with 100% using insulin therapy, and 100% of patients had Wagner-Meggitt’s grade 3 lesions (deep ulcer with abscess or osteomyelitis). Randomisation was by computer generated random numbers. “The randomised allocation sequence, enrolling participants and allocating them to interventions were conducted by a trained staff at the clinic”. While practical for a busy clinic, this would not usually be considered a reliable means of maintaining study blinding.

They report significant decreases in ulcer length (*p* = 0.001), width (*p* = 0.02) and depth (*p* < 0.001) between the groups, however, the study inappropriately reports size to the nearest cm for each dimension (see Figure 1 and Figure 2 in the paper). Data is not provided for success with complete healing. Baseline vitamin D was lower in the vitamin D group, probably non-significantly. As the mean level was deficient, a large proportion of that group were presumably deficient at baseline. A significant reduction in HbA1c compared to the placebo was also reported (*p* = 0.004) as were reductions in LDL and total HDL to cholesterol ratio and C-reactive protein (CRP).

Two prospective cohort studies assessed vitamin D and wound healing [12,30]. Smart et al. (2019) included 80 participants with current foot ulceration who had blood values for vitamin D and HbA1c measured. A three-dimensional camera was used to measure each ulcer. None of the participants included in the study had sufficient vitamin D concentrations, 15% were insufficient and 85% were deficient. The researchers concluded that poor wound healing is linked to vitamin D deficiency and poor glycaemic control using general linear modelling.

The other prospective study included a total of 324 participants, all with diabetes [12]. Of these, 162 had foot ulcers and 162 were ulcer free. Greater vitamin D inadequacy (97.1%) was reported in DFU patients when compared to controls. Low serum vitamin D correlated with foot ulceration by linear regression (*p* < 0.001).

In the study by Pena et al. (2020), it was also reported that 55.7% of patients attending the foot wound clinics in Adelaide were deficient in vitamin D.

These studies indicate potential for vitamin D in the treatment of foot ulcers, particularly in people with baseline deficiency. Replication of the studies in larger cohorts is needed.

#### 3.3.3. Vitamins B12 and Folic Acid

Two studies assessed impact of vitamin B on foot ulceration [27,28]. Badedi et al. (2019) included 323 participants; 108 with DFU and 215 control participants with diabetes. Vitamin B12 concentrations were measured and those with concentrations <148 pmol/L were defined as deficient. Researchers reported a significant correlation between DFU and vitamin B12 deficiency. When covariates were adjusted, there was a significant association between vitamin B12 deficiency and the presence of a DFU, with a 95% CI. Those with diabetes and vitamin B12 deficiency had odds 3.1 times higher of a DFU than those with diabetes without vitamin B12 deficiency. However it has been reported that long term metformin use increases the risk of vitamin B12 deficiency [33]. When compared with placebo, individuals taking metformin had lower vitamin B12 concentrations by 19% [33]. Therefore, this should be taken into consideration for those who have been on metformin for a long period of time.

A non-randomised cohort study by Boykin et al. (2020) recruited participants who were given a daily supplement of 5 mg folic acid (B9), 4 mg cyanocobalamin (B12), and 50 mg pyridoxine (B6) for one month. This study reported significant improvements in ulcer dimensions when comparing the four-week periods before and after supplements (*p* < 0.05).

While further studies are indicated, vitamin B12 deficiency is relatively common in people taking metformin and in people who have a vegan diet. We recommend measuring concentrations in those groups, and treating deficiency whenever it is present.

#### 3.3.4. Vitamin E

In a third study from the Aseni group, Afzali et al. (2019) completed a RCT with 57 participants who were divided into two groups for a 12-week study. Those in the intervention group received 250 mg magnesium oxide plus 400 IU vitamin E daily and the controls received placebo daily. The study reports that magnesium plus vitamin E, compared to placebo, reduced ulcer length (*p* = 0.003), width (*p* = 0.02), and depth (*p* = 0.02). They also reported a significant reduction in HbA1c (*p* < 0.003) and a decrease in triglycerides, LDL-cholesterol, erythrocyte sedimentation rate (ESR) and increase in HDL-cholesterol. It is not clear which part of the co-supplementation led to these positive effects. The study has similar potential issues with measurement of ulcers, and randomization and blinding to the other study discussed above.

#### 3.3.5. Omega-3 Polyunsaturated Fatty Acids

Soleimani et al. (2017) in a 4th study from the Aseni group reported a RCT with 60 subjects; participants were allocated to receive either 1000 mg omega-3 polyunsaturated fatty acids (PUFA) or a placebo twice daily for 12 weeks. By 12 weeks, they reported significant decreases in ulcer length (*p* = 0.03), width (*p* = 0.02) and depth (*p* = 0.01). A significant reduction in HbA1c (*p* = 0.01) and rise in insulin sensitivity (*p* = 0.002) was also reported when compared with the placebo group. Omega-3 PUFA supplementation also showed to significantly decrease CRP levels (*p* = 0.01). The study has similar potential issues with measurement of ulcers, and randomization and blinding.

We note lack of effect of omega-3 PUFA on HbA1c or markers of insulin resistance in people with type 2 diabetes reported in a Cochrane review [34] and in a more recent meta-analysis in 2015 [35].

#### 3.3.6. Minerals (Magnesium, Selenium, Zinc, Copper)

As described above, Afzali et al. (2019) reported testing co-supplementation of magnesium and vitamin E. Magnesium (mg/dL) in the control group was 1.51 at baseline compared to 1.50 at end of study, compared to 1.55 vs. 1.83 (*p* < 0.001) in the intervention group. They reported this had beneficial effects on ulcer size, HbA1c, and lipid concentrations.

In the study by Bolajoko et al. (2017) significant differences in selenium (0.48 vs. 0.81 μmol/L) were noted between the DFU group and the healthy controls (*p* < 0.001). Only non-significant decreases in super-oxide dismutase activity and copper and zinc concentrations were observed (*p* > 0.05).

Pena et al. (2020) measured serum concentrations of copper and zinc in the 131 patients included in the study. Zinc deficiency was present in 26.9% of DFU patients. This study did not perform any statistical analyses for copper as all participants were in the normal range.

Razzaghi et al. (2018) in a fifth study from the Aseni group reported a RCT of 70 participants with DFU. The subjects were randomised to receive either 250 mg of magnesium oxide supplements daily, or a placebo, over a period of 12 weeks. Following the 12-week treatment period, they reported that magnesium supplementation resulted in decreased ulcer length (*p* = 0.01), width (*p* = 0.02), and depth (*p* = 0.003), compared with placebo. The researchers also reported significant reductions in HbA1c (*p* = 0.03) and fasting glucose (*p* = 0.04). Additionally, when compared with the placebo group, there was also a marked reduction in CRP levels (*p* = 0.01). We note that this trial is under investigation by the publishing journal as concerns have been raised about the report.

Momen-Heravi et al. (2017), again from the Aseni group completed a RCT of another cohort of 60 people with DFU who were all also 100% insulin-requiring people who all had 100% Wagner-Meggitt’s Grade 3 ulcers. The participants were randomised to take either 220 mg zinc sulfate, containing 50 mg elemental zinc, or a placebo daily over a period of 12 weeks. The researchers reported that zinc supplementation resulted in significant decreases in ulcer length (*p* = 0.02), width (*p* = 0.02), and depth (*p* = 0.05), when compared to the placebo group. They also reported significant reductions in HbA1c (*p* = 0.01) from baseline to the end of study in the intervention group.

#### 3.3.7. Probiotics

Mohseni et al. (2018) in a sixth study the Aseni group reported evaluating the effects of probiotic supplementation on healing of DFU in 60 participants. Participants were randomly divided into two groups, to receive either a probiotic capsule or placebo daily for 12 weeks. The probiotic contained *Lactobacillus acidophilus*, *Lactobacillus casei*, *Lactobacillus fermentum*, and *Bifidobacterium bifidum*. Once again, they reported significant improvements in ulcer length (*p* = 0.01), width (*p* = 0.02), and depth (*p* = 0.02) in the intervention group, when compared with placebo. Additionally, when compared with the placebo, decreases in total cholesterol (*p* = 0.02) and CRP (*p* = 0.02) levels were also reported. As with all of the other studies from this group, the treatment was reported to cause a significant decrease in HbA1c.

#### 3.3.8. Amino Acids

A RCT by Armstrong et al. (2014) recruited 270 participants with at least one University of Texas grade 1A foot ulcer. Participants were randomised to receive either arginine, glutamine, and beta-hydroxy-beta-methylbutyrate or a placebo, twice daily, for 16 weeks. There were no significant differences in wound healing in non-ischemic patients or those with sufficient baseline albumin concentrations. However, a significantly greater number of those with low baseline albumin concentrations, healed at 16 weeks in the intervention group when compared with controls (*p* = 0.03). Those with an ankle-brachial index of <1.0 also had significant healing rates when compared to the placebo (*p* = 0.008). These analyses are not stated in the report to be pre-specified, so are likely to be post-hoc.

#### 3.3.9. Protein-Energy Supplementation

Eneroth et al. (2004) completed a 6-month RCT of 53 participants with DFU. Participants received either a 400 mL liquid nutritional supplement (20 g protein per 200 mL bottle, 1 kcal/mL, with added vitamins, minerals and trace elements) or a 400 mL placebo daily. Forty patients completed the six-month study and there was no significant effect of supplementation. Baseline protein malnutrition was associated with a non-significantly lower rate of healing at 6 months (24% versus 50% in those without). 

## 4. Discussion

Foot ulceration is a challenging complication seen in patients with diabetes, peripheral neuropathy, abnormal foot architecture and/or peripheral arterial disease. Normal wounds should re-epithelialize within 72 h. Foot ulceration markedly increases the risk of lower limb amputation and can result in prolonged hospitalisations with an increased burden and cost to the healthcare system and the patient [6].

Unfortunately, 6 of the 15 identified studies from the same group (reviewed in Table 5) raise concerns due to homogeneity of the recruited patients and the results from the different treatments, including the report that each of the treatments improved HbA1c and ulcer outcomes significantly. It has also been noted that some of these studies are currently under investigation.

Eight of the other nine studies concluded that concentrations of their studied nutrient correlated with healing or that supplements improved foot ulcer healing. Overall, we conclude that better understanding the mechanism of action of these nutritional supplements, may add to future protocols and guidelines for the treatment of foot ulceration. Most of the sample sizes were small, but the associations are significant enough to suggest that a number of micronutrients may be important in wound healing.

To date, there has only been one similar systematic review completed in 2020 by Moore et al. The authors only included RCTs in their review and reported that it was unclear whether there was a difference in healing or amputations rates between intervention and control groups. Therefore they concluded that there is insufficient evidence to support or refute treating foot ulceration with nutritional interventions and that future studies should be larger and designed to a high standard [36].

Based on the findings of this review, vitamin D and B-complex vitamins may be effective in healing of foot ulceration. Both of these are relatively safe but should be done so under dietetic and/or medical supervision. Supplementation with vitamin C is also considered safe in usual supplemental doses (up to 1000 mg daily) as it is water soluble and excess intake is excreted in the urine [7]. A pilot study available after the date of this systematic review suggests that there is benefit from vitamin C supplements in people with foot ulcers [37].

Depending on the local clinic population, we recommend consideration of testing of appropriate nutritional factors and correction of any deficiencies identified. It is important to note that supplementation of some nutrients can be harmful when the correct dosage is exceeded. Dietary supplementation in the setting of foot ulcer should only be commenced following individual advice, and preferably after measurement of the nutrient under consideration.

### Strengths and Limitations

This is a systematic review, a method which limits bias in determining study inclusion. We deliberately used relatively open search terms, as described in the methods. However, while this review concludes that there are correlations between nutrient deficiencies and foot ulceration, the studies themselves have limitations. As discussed above, six of the articles, which were by the same team of investigators (reviewed in Table 5), raise concerns due to the close similarities between recruited patients and highly similar outcomes across the set of studies. We are unsure whether this data is reliable.

This study did not require HREC/IRB approval as it was a systematic review.

Many of the other studies had small sample sizes and completing further studies using large RCTs will be important. Unfortunately, most of the studies currently published do not report on the two outcomes of greatest clinical interest; complete healing of the ulcer(s) and rate of amputation, and obviously, future studies providing this data is desirable.

Future trials should be of moderate to large scale, preferably multi-centre and look at specific supplementation and the effects on healing of foot ulceration over a period of time. This future research may have the ability to strengthen the evidence that is currently available, and in turn improving outcomes for those with foot ulcers and reducing amputation rates.

## 5. Conclusions

Most published reports find a correlation between nutritional status/supplementation and foot ulcers. However, many of the sample sizes are small, or the studies cannot conclude causation between a specific deficiency and the delay in healing and thus provide associative data. Therefore, further studies should be completed to investigate macronutrient or micronutrient supplementation and foot ulcer healing. Regardless, in clinically appropriate populations, we recommend consideration of testing for nutritional status and replacement in all cases where deficiency is identified.

## Figures and Tables

**Figure 1 nutrients-13-02576-f001:**
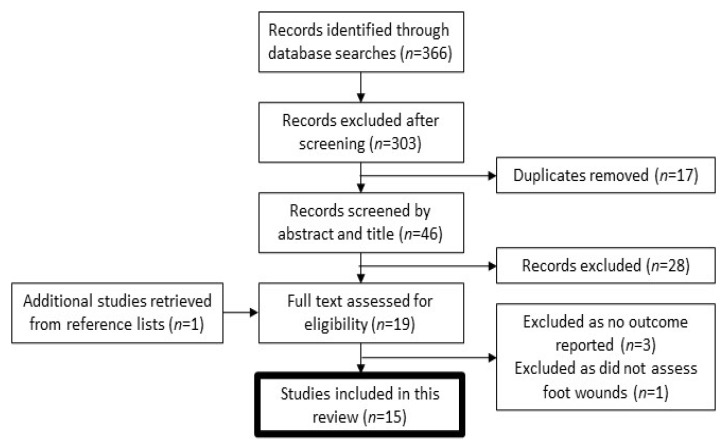
Flow diagram of study selection for the systematic review.

**Figure 2 nutrients-13-02576-f002:**
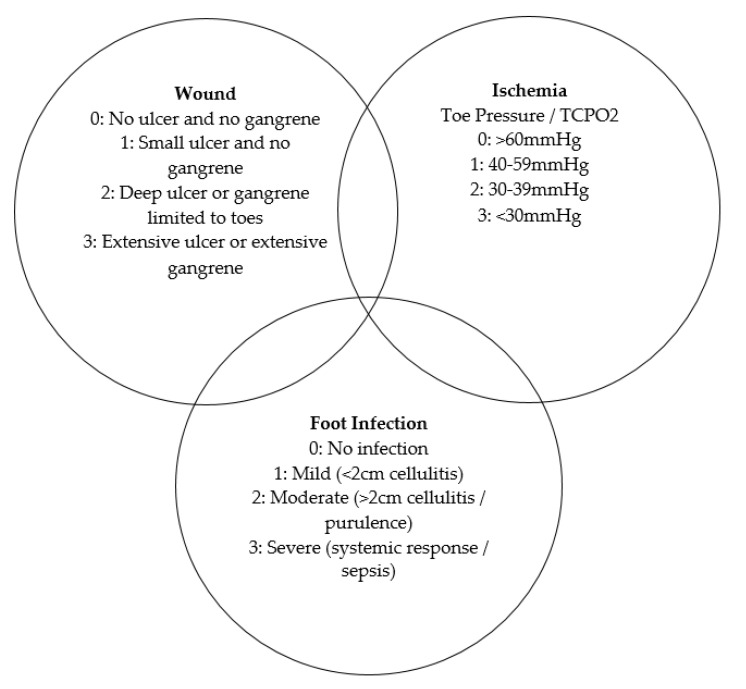
The three intersection rings of the WIfI scoring system to help identify which risk is dominant at a given time [31].

**Table 1 nutrients-13-02576-t001:** Characteristics of the study groups.

Reference, Country	Study Design and Sample Sizes	Age (Years)	Gender (M:F)	BMI (kg/m^2^)	HbA1c (%)	Nutrient(s) or Supplement Studied	Baseline Tested (Y/N)
Afzali et al. Iran	RCT control, *N =* 28	55.5 ± 4.9	22:6	29.7 ± 3.9	7.6 ± 0.6	Magnesium (Mg) and Vitamin E	Y Mg (mg/dL) only, 1.51 ± 0.15 in controls and 1.55 ± 0.18 in intervention
	RCT intervention, *N =* 29	57.2 ± 11.0	23:6	30.3 ± 3.9	7.4 ± 0.8		
Armstrong et al. USA	RCT control, *N =* 141	59 ^	111:30	31.6 ± 7.1	8.0 ± 1.5	Arginine, glutamine, and beta-hydroxy-beta-methylbutyrate	N
	RCT intervention, *N =* 129	58 ^	93:36	33.1 ± 7.3	8.0 ± 1.7		
Badedi et al. Saudi Arabia	Case-control. Control *N =* 215	54 ± 9.8	126:89	28.8 ± 4.3	8.5 ± 2.1	Vit B12	Y Deficiency in 32.1% of controls and 59.2% of cases (*p* = 0.001)
	Cases, *N =* 108	56.9 ± 12.2	66:42	28.9 ± 5.3	10.5 ± 2.0		
Bolajoko et al. Nigeria	Case-control. Control, *N =* 50	51.6 ± 1.03	NR	22.9 ± 0.2	4.1 ± 0.1	Cu, Se, Zn, Vit C and E	Y Significantly lower Vit C (*p* = 0.003), Se and Vit E between case and controls (*p* < 0.001)
	Cases, *N =* 70	51.6 ± 1.07	NR	26.1 ± 0.3	8.6 ± 0.2		
Boykin et al. USA	No controls	-	-	-	-	Folic acid	N
	Retrospective cohort, *N =* 9	67.7 ± 10.2	9:0	NR	8.1 ± 1.4		
Eneroth et al. Sweden	RCT control, *N =* 27	75 ^	21:6	NR	7.0 ± 5.2 ^	400kcal liquid oral supplementation (20 g protein per 200 mL plus other unspecified micronutrients)	Y Protein-energy malnutrition, 44% in placebo group and 19% in intervention
	RCT intervention, *N =* 26	74 ^	19:7	NR	7.1 ± 5.4 ^		
Kamble et al. India	RCT control, *N =* 30	59.7 ± 8.4	23:7	25.9 ± 3.8	9.8 ± 3.5	Vitamin D	Y Vit D (ng/mL) 20.5 ± 9.3 in controls, 18.5 ± 11.6 in intervention
	RCT intervention, *N =* 30	60.2 ± 9.3	25:5	26.4 ± 4.2	9.1 ± 2.1		
Mohseni et al. Iran	RCT control, *N =* 30	58.5 ± 11.0	20:10	25.3 ± 3.7	7.9 ± 0.7	Probiotic	N
	RCT intervention, *N =* 30	62.6 ± 9.7	20:10	26.4 ± 3.0	8.0 ± 0.9		
Momen-Heravi et al. Iran	RCT control, *N =* 30	60.0 ± 10.0	21:9	25.8 ± 3.1	7.9 ± 0.7	Zn	Y Zn (mg/dL) 77.6 ± 12.9 in controls, 76.4 ± 4.5 in intervention
	RCT intervention, *N =* 30	58.3 ± 8.6	21:9	25.8 ± 3.0	7.8 ± 0.9		
Pena et al. Australia	No control group	-	-	-	-	Vit A, C, D; and E, Cu, and Zn	Y Vit A deficiency in 10.9% of patients, Vit D 55.7%, Zn 26.9%. Suboptimal Vit C in 22.2% and deficiency in 50.8%
	Prosp cohort, N= 131	66.3 ± 13.1	104:27	29.4 ± 6.1	8.8 ± 4.4		
Razzaghi et al. (2017) Iran	RCT control, *N =* 30	58.6 ± 8.6	22:8	26.2 ±3.8	7.8 ± 0.7	Vitamin D	Y Vit D (ng/mL) 20.2 ± 15.6 in controls, 15.2 ± 9.9 in intervention
	RCT intervention, *N =* 30	59.6 ± 8.2	22:8	26.0 ± 4.4	8.3 ± 1.0		
Razzaghi et al. (2018) Iran	RCT control, *N* = 35	59.0 ± 10.1	24:11	26.2 ± 4.1	7.8 ± 0.6	Magnesium	Y Mg (mg/dL) 2.0 ± 0.2 in controls and 2.1 ± 0.3 in intervention
	RCT intervention, *N* = 35	60.1 ± 11.1	22:13	28.2 ± 5.2	8.3 ± 1.9		
Smart et al. Bahrain	No control group	-	-	-		Vitamin D	Y Vit D deficiency 85% of cohort, plus insufficient in 15%
	Prosp cohort, *N* = 80	55.8 ± 15.9	57:23	NR	8.2 ± 2.2 (*n* = 72)		
Soleimani et al. Iran	RCT control, *N =* 30	59.9 ± 9.2	23:7	26.9 ± 2.7	7.9 ± 0.7	Omega-3 PUFA from flaxseed oil	N No significant differences in baseline dietary omega-3 intake
	RCT intervention, *N =* 30	58.8 ± 11.2	23:7	27.0 ± 4.5	7.5 ± 1.5		
Zubair et al. India	Prosp cohort control, *N =* 162	47.1 ± 12.1	102:58	24.0 ± 4.2	7.9 ± 0.9	Vitamin D	Y Vit D (ng/mL) 29.8 in controls and 8.4 in cases (*p* < 0.005)
	Prosp cohort cases, *N =* 162	46.3 ± 13.2	103:59	24.8 ± 4.5	9.6 ± 2.0		

Data are presented as mean ± standard deviation, with decimals rounded to 1 place. NR = not reported. RCT = randomised controlled trial. Pros*p* = prospective. Mg = magnesium, Vit = vitamin, Cu = copper, Se = selenium, Z*n* = zinc, PUFA = polyunsaturated fatty acids. ^ = median. Other data show mean ± SD.

**Table 2 nutrients-13-02576-t002:** Wagner-Meggitt classification.

Stage/Grade	
0	No open lesions; may have deformity or cellulitis
1	Superficial diabetic ulcer (partial or full thickness)
2	Ulcer extension to ligament, tendon, joint capsule, or deep fascia without abscess or osteomyelitis
3	Deep ulcer with abscess, osteomyelitis, or joint sepsis
4	Gangrene localized to portion of forefoot or heel
5	Extensive gangrenous involvement of the entire foot

**Table 3 nutrients-13-02576-t003:** University of Texas foot ulcer classification.

Stage/Grade	0	1	2	3
A	Pre or post ulcerative lesion completely epithelialised	Superficial wound. Not involving tendon, capsule, or bone	Wound penetrating to tendon or capsule	Wound penetrating to bone or joint
B	With infection	With infection	With infection	With infection
C	With ischemia	With ischemia	With ischemia	With ischemia
D	With infection and ischemia	With infection and ischemia	With infection and ischemia	With infection and ischemia

**Table 4 nutrients-13-02576-t004:** Overview of study design and results of included articles other than the Aseni group.

Author	Study Design	Primary Measure	Exclusion Criteria	Results	Micronutrients (Basal→Final)	Limitations
Armstrong et al.	RCT	Arginine, glutamine and beta-hydroxy-beta-methylbutyrate	Pregnancy, <6 weeks post-partum, breastfeeding, <18 years old, ulceration on lesser digits that was diabetic or neuropathic in aetiology, ulcer <30 days or > 12 months duration, ulcer surface area <1 cm^2^ or >10 cm^2^, ankle-brachial index <0.7 or >1.2, change in medication during the trial period, any dietary supplements or alternative therapies, not agreeable to wear offloading device	No difference in wound closure or time to wound healing in non-ischemic patients or those with normal baseline albumin. Post-hoc analysis of those in the intervention group with baseline low albumin levels showed improved healing at 16 weeks vs. placebo (*p* = 0.03). Those with ABI <1.0 had increased healing rates when compared to placebo (*p* = 0.008).	N/A	Study limited to those with University of Texas ulcer classification 1A. Study period of 16 weeks which may not be long enough to identify overall benefit in larger population group
Badedi et al.	Case-control	Vit B12	Anyone taking B12 supplementation	B12 deficiency was significantly associated with DFU (odds ratio 3.1), indicating patients with vitamin B12 deficiency were 3 times more likely to develop a foot ulcer	N/A	Diabetes duration was significantly longer in DFU group, and higher rate of neuropathy and arterial disease. The study design cannot prove causation.
Bolajoko et al.	Case-control	Vit C and E, Cu, Se and Zn	Pregnancy, healthy controls with fasting glucose >5.6 mmol/L, peripheral arterial disease, osteomyelitis at ulcer site, those with renal or liver disease	Significantly lower vitamin C (*p* = 0.003), selenium and vitamin E in those with DFU vs. controls (*p* < 0.001). No significant change in copper or zinc	N/A	Control group was not diabetic. Ulcer size was not correlated with vitamin levels
Boykin et al.	Retro cohort	5 mg folic acid, 4 mg cyanocobalamin (B12), and 50 mg pyridoxine (B6)	If NPWT had been used previously, any change in medication within 1 month of folic acid treatment, or the wound had reduced in size by 50% 4 weeks prior to starting folic acid	Significant improvements in wound areas for 4-week periods before and after high dose folic acid treatment (*p* < 0.05)	N/A	Small sample size, blood tests not completed prior to treatment with high dose folic acid, no control group
Eneroth et al.	RCT	400 kcal liquid oral supplementation (20 g protein per 200 mL plus other unspecified micronutrients)	Active chronic inflammatory intestinal disease, immunosuppressive treatment, malignancy, decreased kidney function, severe heart disease, psychiatric or addictive illness	At 6 months wound healing achieved in 8 out of 23 patients (41%) in the placebo group, and in 7 out of 17 (35%) in the intervention group. 24% of patients with protein energy malnutrition had healed at 6 months when compared with 50% of those without it. Neither of these results were significant.	Y Protein-energy malnutrition, 44%→52% of those in placebo group and 19%→27% in the intervention	Small sample size, did not assess a specific nutrient, regular food and fluid intake was not assessed after the intervention, not known if the supplement led to a decrease in normal food or fluid intake
Kamble et al.	RCT	60,000 IU Vit D weekly	Non diabetic foot ulcer, chronic kidney disease, liver disease, taking immunosuppressant’s or calcium supplements	Decrease in HbA1c (*p* = 0.008) and total cholesterol (*p* = 0.05) vs. controls. Decrease in wound surface area (*p* = 0.0001)	Y Baseline Vit D 20.5→20.1 ng/mL in controls and 18.5→31.0 (*p* = 0.0001) in D group	Small sample size, wound area was measured by calculation of greatest length and width, not using digital photography or wound analysis software
Pena et al.	Prosp cohort	Vit A, C, D; and E, Cu and Zn	Nondiabetic, under 18 years of age	Increased severity of DFU associated with lower vitamin C concentrations (*p* = 0.02). 1 in 5 patients had non-measurable vitamin C	N/A	Clinical correlation not assessed with healing outcomes. Further research needed to identify clinical implications of deficiencies and effect on healing
Smart et al.	Prosp cohort	Vit D	Anyone under 18 years of age	Poor wound healing associated with older age and higher HbA1c (*p* < 0.0001). Exposed bone, or temperature difference over 3°F were linked to poor healing (*p* < 0.006). Lower 25(OH)D levels correlated with poor healing	N/A	No control group. Only a cross-sectional study and a RCT follow up to assess causal link between vitamin D and wound healing outcomes
Zubair et al.	Prosp cohort	Vit D	Patients with inflammatory, infectious, autoimmune, rheumatic diseases, cancer, or severe renal or liver failure. Those taking anti-inflammatory drugs. Those with recent VTE	Higher vitamin D inadequacy (97.1%) in DFU patients vs. diabetic controls. HbA1c, triglycerides, neuropathy, retinopathy, hypertension, smoking and nephropathy were all linked to DFU development		Diabetic subjects with and without foot disease included. Unclear whether vitamin D is directly related to delayed wound healing or a secondary effect

RCT = randomised controlled trial. ABI = ankle brachial index. Retro = retrospective, Pros*p* = prospective. Vit = vitamin, Cu = copper, Se = selenium, Z*n* = zinc. NPWT = negative pressure wound therapy. VTE = venous thromboembolism.

**Table 5 nutrients-13-02576-t005:** Reports from the Aseni group.

First Author, City, Year	Nutrient(s) Tested	Sample Size Initial→Final	Age (y)	M:F	BMI (kg/m^2^)	Ins Rx	W-M Grade	HbA1c Basal→Final (%)	Length Basal→Final (cm)	Width Basal→Final (cm)	Depth Basal→Final (cm)	Micro-Nutrients Basal→Final	Allocation and Notes
Afzali ## Kashan 2019	250 mg Mg oxide + 400 IU Vit E	30 placebo→28	55.5 ± 4.9	22:6	29.7 ± 3.9	NR	3	7.6→7.4	3.1→2.3	2.5→1.8	1.1→0.9	1.51→1.50	Block randomised, 2 tablets from different makers for active group, unclear if same number of tablets for placebo group (third manufacturer).
		30 active→29	57.2 ± 11.0	23:6	30.3 ± 3.9	NR	3	7.4→6.8 **	2.8→1.6 *	2.1→1.2 *	0.9→0.4 *	1.55→1.83 (*p* < 0.001)	
Mohseni ## Babol 2018	Probiotic 2 × 10^9^ CFU/g each	30 placebo→28	58.5 ± 11.0	20:10	25.3 ± 3.7	100%	3	7.9→7.7	3.2→2.4	2.6→1.9	1.1→0.8	N/A	Randomised by clinic staff. 1 placebo subject lost to follow-up but paper states 30 people analysed. Not clear where data has fewer subjects.
		30 active→30	62.6 ± 9.7	20:10	26.4 ± 3.0	100%	3	8.0→7.4 **	3.2→1.9 *	2.4→1.3 *	1.2→0.7 *		
Momen-Heravi ## Kashan 2017	220 mg zinc sulphate	30 placebo→28	60.0 ± 10.0	21:9	25.8 ± 3.1	100%	3	7.9→7.8	3.1→2.2	2.7→1.9	1.3→1.0	77.6→74.0	Randomised by clinic staff. 2 placebo subjects lost to follow-up and 30 people analysed. Not clear where data has fewer subjects.
		30 active	58.3 ± 8.6	21:9	25.8 ± 3.0	100%	3	7.8→7.3 **	3.1→1.6 *	2.9→1.5 *	1.3→0.8 *	76.4→89.1 (*p* < 0.001)	
Razzaghi ^^ Kashan 2017	50,000 IU Vit D	30 placebo→28	58.6 ± 8.6	22:8	26.2 ± 3.8	100%	3	7.8→7.7	NR. −1.1 cm	NR −1.1 cm	NR −0.5 cm	20.2→18.4	Randomisation by clinic staff. 2 placebo subjects lost to follow-up and 30 people analysed. Unclear where data has lower N. Ulcer size in 1cm increments.
		30 active	59.6 ± 8.2	22:8	26.0 ± 4.4	100%	3	8.3→7.7 **	NR −2.1 cm *	NR −2.0 cm *	NR −1.0 cm *	15.2→28.1 (*p* < 0.001)	
Razzaghi ~~ Kashan 2018	250 mg Mg oxide	35 placebo→31	59.0 ± 10.1	24:11	26.2 ± 4.1	100%	3	7.8→7.7	3.6→2.7	2.9→2.1	1.3→0.9	2.0→1.9	Randomised by clinic staff. 9 lost to follow-up but reported as 70 people analysed. Not clear where data has fewer subjects.
		35 active→30	60.1 ± 11.1	22:13	28.2 ± 5.2	100%	3	8.3→7.6 **	3.6→1.8 *	3.3→1.7 *	1.7→0.9 *	2.1→2.3 (*p* < 0.001)	
Soleimani ^^ Kashan 2017	1000mg omega-3 PUFA from flaxseed bd	30 placebo→28	59.9 ± 9.2	23:7	26.9 ± 2.7	100%	3	7.9→7.8	3.4→2.4	2.9→1.9	1.3→0.8	N/A	Randomised by clinic staff. 5 lost to follow-up, reported as 60 analysed. Unclear where N is lower.
		30 active→27	58.8 ± 11.2	23:7	27.0 ± 4.5	100%	3	7.5→6.6 **	3.5→1.4	2.9→1.1	1.4→0.5		

M:F = male and female participants. Ins Rx = insulin therapy. W-M = Wagner-Meggitt’s ulcer grade. NR = information not reported in the paper. Mg = magnesium. Vit = vitamin. Data was presented in the papers as mean ± standard deviation. *, ** = statistically significant in the report. ## Published in Wiley journal ^^ Published in an Elsevier journal ~~ Published in Springer Link Journal.

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
