# Peer review of "Associations between Nutrients and Foot Ulceration in Diabetes: A Systematic Review"

_nutrients, 2021, doi:10.3390/nu13082576_

Round 1
Reviewer 1 Report
Dear authors, after reviewing the article, I made the following indications:
1. The article ensures that the message you want to convey reaches the reader clearly and directly.
2. I have verified that each section of the manuscript has the necessary and timely information.
3. The line of argument is verified, that is, the coherent sequence and the logical and complete reasoning.
4. The methodology is reviewed according to a systematic review.
Author Response
Thank you for the comments.
We have checked the paper and made some further corrections.
Reviewer 2 Report
I was honored to review the manuscript entitled: Associations between nutrients and foot ulceration in diabetes: a systematic review submitted to Nutrients. The study presents high quality and deals with the important clinical issues, such a type of study is needed. I have only a few small remarks that authors should address properly.
I recommend accepting the manuscript after minor revision.
There are only some points to correct:
- please provide the list of abbreviations
- introduction and discussion section need improvement; please provide information on how your results will translate into clinical practice
- in the discussion section please provide study strong points and study limitation section
- please correct typos
All the abovementioned issues are crucial for the credibility of the results. The paper can be accepted only after addressing all the issues and another subsequent review.
Author Response
Thank you.
We have added an abbreviation list to the end of the abstract, and added a subheading in the final section to separate strengths and limitations, and expanded upon that area.
We have all proof-read the document and have made some further corrections.
This manuscript is a resubmission of an earlier submission. The following is a list of the peer review reports and author responses from that submission.
Round 1
Reviewer 1 Report
This metanalysis of the effects of micronutrient supplementation in healing of diabetic foot ulcers addresses an important issue.
Major comments:
Figure 1 is jumbled up and needs reformatting.
The results section is hard to read because of mixing up of comments on the groups of studies and outcomes. These could be clarified by separating the comments from results with the latter including significances in a separate Table or streamlined table 6
I was not clear how many of the studies actually recorded baseline serum levels of the micronutrients studied. An analysis of outcomes where low levels of micronutrients were corrected would be important for this review and for suggested future studies.
Author Response
Thank you for the invitation to revise our paper “Associations between micronutrient levels and foot ulceration in diabetes: a systematic review”. Please find the revision attached, with changes tracked.
We have made the changes as requested by both reviewers. This included reformatting figure 1, removing subheadings in the introduction and removing bullet points in methods and results. In table 6, we bolded any significant p values, we didn’t report those in table 5 as we had concerns about all studies being from the same group with at least one paper currently under formal investigation.
We have added a column to table 1 to indicate whether baseline serum levels specific to the study were recorded. If they were measured, we noted the rate of deficiency in the population or the baseline levels in the case and control groups. We then added a column in tables 5 and 6 with a comparison of baseline serum levels to levels at the end of the study if applicable.
Thank you for your consideration of this manuscript.
Reviewer 2 Report
The present study summarized the existing evidence on micronutrient intake and foot ulceration in diabetes. The study seems complete and comprehensive. The authors should amend few minor points before considering the paper for publication.
- I would avoid subheadings in the introduction. Since this paper is presented as a systematic review, I would suggest to simply discuss about the vitamins listed without headings.
- The bullet point list in materials and methods (as well as in the results) is not a proper style for information presentation in a scientific paper. I suggest the authors to provide a normal list, perhaps numbered in roman numbers, such as (i) …; (ii) …; (iii) ….; (iv) …. And so on.
- In table 1, it is not described what supplementation has been experimented in the study of Eneroth et al.
Author Response

(The authors gave the same response as above.)

Reviewer 3 Report
This review purports to assess the ‘associations between micronutrient levels and foot ulceration in diabetes’. However, many of the studies reported do not assess the micronutrient status of their cohorts, nor do they actually supplement with micronutrients (vitamins and trace minerals). I get the feeling the authors of this review set out to investigate the role of vitamin C in diabetic foot ulcers (based on their search terms), but there were insufficient studies, so they included other so-called ‘micronutrients’ (probiotics are not nutrients, they are microorganisms). This review is not acceptable in its current form and needs a complete rethink and rewrite based on my comments below.
Line 19 – why are only vits C and D mentioned in the abstract when other micronutrients were investigated?
Lines 32 & 34 – make it clear if referring only to Australian stats or to global stats. If only Australian stats then would be good to also include global stats to make the review more generalisable.
It is not clear that the authors are looking specifically for studies that measured the micronutrient status of their cohorts – this should really be specified as an inclusion criteria. Table 1 looks like outcomes of micronutrient interventions were what was of interest.
Could assessment of study quality be included eg using the Cochrane Risk of Bias tool.
Table 1. Micronutrients are specifically vitamins and trace minerals. The authors have included studies that did not administer micronutrients eg Arginine, glutamine and beta-hydroxy-beta-methylbutyrate, probiotic, Omega-3 PUFA from flaxseed oil. Also, not clear what is in the 400kcal liquid oral supplementation.
Did any of the studies in Table 1 measure micronutrient levels (as not clear from the table and that is the title of the review)? Could the reference numbers also be included in Table 1 for easy reference.
Tables 2 & 3 should be in methods or appendix or supplementary material - not in results.
Figure 2 – writing within the circles is too small to read easily – increase the font size. Also, this should not be in the results, but instead within methods, appendix or supplementary material.
Line 195 – why are concentrations of vitC not included, only P value? (I see later there is discussion regarding potential issues with the assay) And why are P values not included for other micronutrients?
Why are multinutrients discussed under the vitamin C heading?
Line 288 - Omega-3 polyunsaturated fatty acids are not ‘micronutrients’. If there is only one study, and it is from a dubious source, then maybe this can be left out altogether?
Line 302. Did they measure magnesium levels in the participants before and after supplementation? If not, then maybe not include this?
Line 304 - significant differences in selenium – what – concentrations?
Line 318. If the Aseni group studies did not measure the micronutrient status of their participants then this would justify leaving them out of this review, particularly if ‘this trial is under investigation by the publishing journal as concerns have been raised about the report.’ There is not point reporting on studies that are dubious and could possibly be retracted. It sounds like they are reporting very similar results regardless of the interventions given, which is also improbable.
Line 329. Probiotics are not ‘micronutrients’. This should be excluded from the review – particularly since also from the dubious Aseni group.
Line 341 – amino acids are not ‘micronutrients’.
Line 353 – Energy or protein supplementation?
Table 5. If this group did not include measures of micronutrient status of their cohorts (or did not supplement with micronutrients) then could be excluded from the review (particularly since all their studies seem questionable).
Table 6. Results should include the baseline and post-supplementation micronutrient status.
Discussion will need a rewrite based on the changes suggested above.
Line 412 – it is not until the conclusions that the authors introduce the term ‘macronutrients’. This should be introduced earlier if macronutrient studies are being included along with the micronutrient studies.
Minor
Line 57 – delete do from ‘otherwise do not’
Line 62 - neutrophil ‘oxidative burst’
Figure 1 – reasons for exclusions usually included in flow diagram
Lines 140-142 – delete these sentences as instructions.
Line 144 & 146 &153 – use past tense – were not are, was not is, had not have
Line 193 - oxidative stress
Line 196 - were found not was found
Line 203 – include Pena reference number
Line 252 - possibly particularly in people – pick either possibly or particularly
Throughout – leave spaces between numbers and units
Throughout – the term ‘concentrations’ should be used instead of ‘levels’